

# Effects of mixing on resolved and unresolved scales on stratospheric age of air

Dietmüller Simone[1], Garny Hella[1], Plöger Felix[2], Jöckel Patrick[1], and Cai Duy[1]

[1]Deutsches Zentrum für Luft- und Raumfahrt (DLR), Institut für Physik der Atmosphäre, Oberpfaffenhofen, Germany
[2]Institute of Energy and Climate Research (IEK-7), Forschungszentrum Jülich GmbH, Jülich, Germany

*Correspondence to:* S. Dietmüller (simone.dietmueller@dlr.de)

**Abstract.**

Mean age of air (AoA) is a widely used metric to describe the transport along the Brewer-Dobson circulation. We seek to untangle the effects of different processes on the simulation of AoA, using the chemistry-climate model EMAC and the Lagrangian chemistry transport model CLaMS. Here, the effects of residual transport and two-way mixing on AoA are calculated. To do so, we calculate the residual circulation transit time (RCTT). The difference of AoA and RCTT is defined as aging by mixing. However, as diffusion is also included in this difference, we further use a method to directly calculate aging by mixing on resolved scales. Comparing these two methods of calculating aging by mixing allows for separating the effect of unresolved aging by mixing (which we term "aging by diffusion" in the following) in EMAC and CLaMS. We find that diffusion impacts AoA by making air older, but its contribution plays a minor role (order of 10%) in all simulations. However, due to the different advection schemes of the two models, aging by diffusion has a larger effect on AoA and mixing efficiency in EMAC, compared to CLaMS. Regarding the trends in AoA, in CLaMS the AoA trend is negative throughout the stratosphere except in the northern hemisphere middle stratosphere, consistent with observations. This slight positive trend is neither reproduced in a free-running nor in a nudged simulation with EMAC - in both simulations the AoA trend is negative throughout the stratosphere. Trends in AoA are mainly driven by the contributions of RCTT and aging by mixing, whereas the contribution of aging by diffusion plays a minor role.

## 1 Introduction

The large-scale Brewer-Dobson circulation affects the chemical composition of the stratosphere, as it describes all transport processes of an air parcel on its way through the stratosphere, including both the mean mass transport along the residual circulation and the two-way exchange of air mass, referred to as mixing. Mean Age of Air (AoA) is a common measure to quantify the overall capabilities of a global model to simulate stratospheric transport. It is defined as the mean transport time of an air parcel from the entry region at the tropical tropopause to any region in the stratosphere (Hall and Plumb, 1994; Waugh and Hall, 2002). With the concept of AoA stratospheric mass transport can also be derived from observations of inert tracers as e.g. $SF_6$ or $CO_2$ (e.g. Andrews et al., 2001; Engel et al., 2009; Stiller et al., 2012) and can be directly compared to chemistry-climate models (CCMs).



Model data inter-comparisons of simulated AoA in stratosphere-resolving climate chemistry models (CCMs) (see e.g. Hall et al., 1999; Eyring et al., 2006; Butchart et al., 2010; SPARC CCMVal, 2010), showed large spread between the models with most CCMs having a significantly lower AoA than derived from in-situ observations (Andrews et al., 2001; Engel et al., 2009). The CCMVal-2 (SPARC CCMVal, 2010) model inter-comparison, which is using the data of 15 CCMs, reported for 7 of 15 models a good agreement of AoA at 50 hPa with observations, also their tropical AoA profiles are within the uncertainties of observations at all altitudes. However, for most of these models the spread of mid-latitude AoA is significant and AoA is too low in the middle stratosphere, if compared to in-situ observations (see SPARC CCMVal, 2010, their figure 5.5).

Regarding the trends of AoA, current CCMs show negative trends throughout the stratosphere due to strengthening of the residual circulation (leading to shorter mean transport time) in a warmer climate. This is a well known feature in current CCMs (e.g. Garcia and Randel, 2008; Butchart et al., 2010; Butchart, 2014). In contrast, the estimates of the longest existing observationally based AoA data set from balloon flights shows an insignificant weakly positive trend in the northern hemisphere mid-latitudes for the last 30 years (see Engel et al., 2009). Moreover, observations with the Michelson Interferometer for Passive Atmospheric Sounding (MIPAS) instrument exist for the years 2002-2012 (Stiller et al., 2012) and they are showing mainly a decrease in AoA in the southern hemisphere and an increase in the northern hemisphere. Chemical Transport Models (CTMs), if driven by certain reanalysis data, are able to qualitatively reproduce the observationally based AoA trends of Engel et al. (2009) and Stiller et al. (2012), as shown by Monge-Sanz et al. (2013); Ploeger et al. (2015a); Diallo et al. (2012).

A better understanding of the processes that control AoA is crucial to understand the model spread in AoA and to reconcile current discrepancies between simulated and observed long-term changes in AoA. Here it is important to quantify, besides the effect of mean transport along the residual circulation, the effect of eddy mixing (in the following defined as "mixing") on AoA. Increase in mixing causes a strengthening in recirculation, and an increase in AoA (Neu and Plumb, 1999). Garny et al. (2014) investigated the effect of mixing on AoA and found out that mixing makes air older throughout most parts of the lower stratosphere, except in the extratropical lowermost stratosphere, where mixing reduces AoA. However, they did not exactly calculate aging by mixing on resolved scales, as they defined aging by mixing as the difference between simulated AoA and the transit time along the residual circulation. This difference also includes aging by mixing on unresolved scales, as AoA in global models is also affected by parametrized and numerical diffusion (Garny et al., 2014). Recently, Ploeger et al. (2015b) explicitly investigated aging by mixing on resolved scales, by integrating the exact calculated local mixing tendencies along the trajectories of the residual circulation. In agreement with Garny et al. (2014), they also showed that mixing significantly increases AoA, except in the lower polar stratosphere. Moreover, Ploeger et al. (2015b) investigated the effects of mixing and residual circulation on AoA (trends) with a CTM, driven by European Center for Medium-Range weather Forecast ERA-Interim reanalysis data (Dee et al., 2011). They found, that for 1990-2013 AoA decreases in most of the lower stratosphere, largely caused by the effect of aging by mixing.

Differences in the numerical formulation of a model could contribute to the model spread in AoA trends. Eluszkiewicz et al. (2000) showed that AoA (simulated in one model) is very sensitive to the advection algorithm used to integrate the tracer continuity equation. However, Eyring et al. (2006) compared transport properties between different models and they came to the result that there is little difference in key transport diagnostics between models with spectral and flux-form advection schemes





(Shepherd, 2007). Also the choice of the vertical coordinate (pressure or potential temperature) may influence the AoA pattern (e.g. Mahowald et al., 2002; Hoppe et al., 2016). The recent work of Hoppe et al. (2016) investigated the differences in AoA of two CCM simulations with the same underlying model, but one using a flux-form semi-Lagrangian scheme and corresponding kinematic vertical velocities and another simulation using a Lagrangian scheme and diabatic vertical velocities. They found out, that the difference pattern of AoA can be attributed both to the different vertical velocities and to the different transport schemes, leading to differences in aging by mixing. In particular in regions of strong transport barriers, like the polar vortex, the Lagrangian simulation has been shown to result in more realistic transport characteristics (Hoppe et al., 2014).

In this study we quantify the effects of different processes that affect the simulation of AoA. We focus on the effect of aging by mixing on resolved scales and on unresolved scales. To do so we use simulations with the chemistry climate model system EMAC (ECHAM/MESSy Atmospheric Chemistry) and with the Chemical Lagrangian Model of the Stratosphere CLaMS. Note that the two models differ in the advection schemes and that they have different contributions from unresolved diffusion. A brief description of models and simulation setups will be given in Section 2. We summarize the methods for separating the effects on AoA in Section 3. Annual zonal mean climatologies of all processes affecting AoA (effect of residual circulation, effect of mixing processes both on resolved and unresolved scales) are given in Section 4. Moreover, the differences between the different model simulations, as well as sensitivity experiments, are discussed there. In Section 4.4 the long-term trends of AoA, RCTT and mixing are investigated for all simulations and the model differences will be discussed. Conclusions are given in Section 5.

## 2 Model Simulations

### 2.1 Model description of the chemistry climate model EMAC

The numerical chemistry climate model system EMAC (ECHAM/MESSy Atmospheric Chemistry) includes sub-models describing tropospheric and middle atmosphere processes and their interaction with ocean, land, and human influences (Jöckel et al., 2010). It uses the second version of the Modular Earth Submodel System (MESSy2) to link different submodels for physical and chemical processes in the atmosphere (Jöckel et al., 2010). The core atmospheric model of EMAC is the 5th generation of the European Centre Hamburg general circulation model ECHAM5 (Roeckner et al., 2006). Atmospheric tracer management in MESSy is treated with the submodel TRACER (Jöckel et al., 2008), providing an interface structure (memory and data management) to couple chemical processes with the basemodel. In the standard setup of EMAC tracers are transported by the flux-form semi-Lagrangian (FFSL) transport scheme of Lin and Rood (1996). EMAC employs a hybrid pressure grid structure and vertical (kinematic) velocities for tracer transport are calculated internally by the transport scheme as a residual from the horizontal flux divergence using the continuity equation (Lin, 2004; Lauritzen et al., 2011). Furthermore, transport by vertical diffusion is parametrized (Brinkop and Roeckner, 1995).





### 2.1.1 Model description of the Chemical Lagrangian Model of the Stratosphere CLaMS

The Lagrangian chemistry transport model CLaMS (Chemical Lagrangian Model of the Stratosphere) combines forward trajectories with parametrized small-scale mixing. Small-scale mixing is implemented in a physical manner, such that mixing is induced by deformations in the large-scale flow. The model uses an isentropic vertical coordinate (potential temperature) throughout the stratosphere, with the cross-isentropic vertical velocity deduced from the total diabatic heating rate, including all-sky radiative, latent and turbulent heating contributions (here taken from ERA-Interim reanalysis, see Sect. 2.2). Further details of the model set-up used here can be found in Pommrich et al. (2014).

A particular advantage of CLaMS Lagrangian transport is that the trajectory calculation is non-diffusive per se, and that the strength of diffusion induced by parameterized small-scale mixing may be controlled. For that reason, a critical Lyapunov exponent $\lambda_c$ has to be specified, which controls the relative distance between model gridpoints to be affected by mixing (for details see e.g. Hoppe et al., 2014).

## 2.2 Model simulations with EMAC and CLaMS

Table 1 gives an overview over all model simulations used for the present study. A detailed description of these listed simulations will be given within this section.

In the Earth System Chemistry integrated Modelling ’(ESCiMo)’ initiative (Jöckel et al., 2016), reference simulations (RC) as defined by the IGAC/SPARC Chemistry-Climate Model Initiative (CCMI) and described in detail by Eyring et al. (2013) were performed. In our study we focus on two of these ESCiMo reference simulations (namely RC1-base-07 and RC1SD-base-07), both conducted in the T42L90MA resolution. This resolution has a spherical truncation of T42 (corresponding to a quadratic Gaussian grid of approx. 2.8 by 2.8 in latitude and longitude) and a vertical resolution of 90 hybrid pressure levels with the uppermost level centered at 0.01 hPa.

The first simulation we use is RC1-base-07 (in the following referred to EMAC-RC1), a free-running hindcast simulation, ranging from 1960 to 2011. The sea surface temperatures (SSTs) and the sea ice concentrations (SICs) are used from the HADISST data base, provided by the UK Met Office Hadley Centre (available via http://www.metoffice.gov.uk/hadobs/hadisst/). The second simulation we use is RC1SD-base-07 (in the following referred to EMAC-RC1SD), a hindcast simulation with specified dynamics (SD), ranging from 1980 to 2011. Nudging is done by a Newtonian relaxation technique towards 6 hourly ECMWF reanalysis data (ERA-Interim, Dee et al. (2011)). The nudging of the prognostic variables, divergence, vorticity, temperature and the (logarithm of the) surface pressure is applied in spectral space with a corresponding relaxation time of 48, 6, 24 and 24 h, respectively. Global mean temperature is also included. Nudging is applied in the troposphere from above the boundary layer up to 5 hPa, with nudging coefficients decreasing with height above 10 hPa (for details see Jöckel et al. (2016)). Nudging further implies that SSTs and SICs are used from ERA-Interim reanalysis data.

For the simulations of this paper the transport model CLaMS was driven with meteorological data from ERA-Interim reanalysis over the period 1990–2011. Cross-isentropic vertical velocity has been deduced from the reanalysis forecast total diabatic heating rate (see Pommrich et al., 2014). We carried out a high-resolution reference simulations (CLAMS-ERAI), with the



**Table 1.** Overview over the model simulations with EMAC and CLaMS, used for the present study. The simulations differ with respect to dynamics, tracer transport (advection scheme and for CLaMS also the mixing strength, expressed by a critical Lyapunov exponent $\lambda_c$ in $[\text{day}^{-1}]$) and resolution.

| simulation | analyzed years | dynamics | tracer transport | resolution |
| --- | --- | --- | --- | --- |
| EMAC-RC1 | 1990-2011 | free running | FFSL | T42L90MA |
| EMAC-RC1SD | 1990-2011 | nudged to ERA-Interim | FFSL | T42L90MA |
| CLAMS-ERAI | 1990-2011 | driven by ERA-Interim | Lagrangian, $\lambda_c = 1.5$ | $\approx 100\text{km}$ |
| CLAMS-L1.5 | 1990-2010 | driven by ERA-Interim | Lagrangian, $\lambda_c = 1.5$ | $\approx 200\text{km}$ |
| CLAMS-L1.0 | 1990-2010 | driven by ERA-Interim | Lagrangian, $\lambda_c = 1.0$ | $\approx 200\text{km}$ |

critical Lyapunov exponent chosen as $\lambda_c = 1.5$ day$^{-1}$, resulting in good agreement with observed trace gas distributions as shown in several recent publications (e.g., Pommrich et al., 2014; Ploeger et al., 2015a). Furthermore, for the investigation of model diffusion effects we carried out two low-resolution sensitivity simulations, both driven by ERA-Interim meteorology but with varying the strength of parametrized small-scale mixing (either $\lambda = 1.5$ day$^{-1}$ or $\lambda = 1.0$ day$^{-1}$). The sensitivity

simulation with $\lambda = 1.5$ day$^{-1}$ (CLAMS-L1.5) is close to the ERA-Interim reference simulation (only difference is horizontal resolution) and the simulation with $\lambda = 1.0$ day$^{-1}$ (CLAMS-L1.0) includes enhanced parametrized mixing causing stronger diffusion.

## 3  Calculating AoA, residual transport, mixing and diffusion

### 3.1  Calculation of AOA

As mentioned above, stratospheric mean age of air is defined as the residence time of air parcels in the stratosphere (e.g Hall and Plumb, 1994; Waugh and Hall, 2002). It is affected both by the residual circulation and by eddy mixing. In global models an AoA tracer is implemented as an inert tracer with linearly increasing boundary conditions ("clock-tracer"; Hall and Plumb (1994)). AoA at a certain grid point in the stratosphere is then calculated as the time lag between the local tracer mixing ratio (at this certain grid point) and the current mixing ratio at a reference point. In the EMAC simulation setup AoA is obtained

from linearly increasing mixing ratios of an inert synthetic tracer (Age of air tracer, see table A1 in Jöckel et al. (2016)). The AoA tracer is emitted into the lowermost model level. The reference point is set at the tropical tropopause for age tracer mixing ratios between 10°S and 10°N as the height of thermal tropopause.

In CLaMS there is an analogous AoA tracer emitted into in the lowest model layer. Mean age in the stratosphere is calculated as the time lag between the local tracer mixing ratio and the mixing ratio at the boundary layer. To be consistent with EMAC the

AoA value at 340 K between 10°S and 10°N (corresponding approximately the height of the tropical tropopause) is subtracted from AoA.





## 3.2 Calculation of residual circulation transit time

The residual circulation transit time (RCTT) is the hypothetical age, air would have if it was only transported by the residual circulation, without eddy mixing. For the EMAC simulation data RCTTs are calculated following Birner and Bönisch (2011) by calculating backward trajectories that are driven by the Transformed Eulerian Mean (TEM) meridional and vertical monthly

velocities (referred to as residual velocities) with a standard fourth-order Runge-Kutta integration. The backward trajectories are initialized on a grid with 64 latitudes and 42 pressure levels (from 200 to 5 hPa). The residual meridional velocity $\overline{v^*}$ and the vertical velocities $\overline{w^*}$ are calculated following Andrews et al. (1987) with data from 6-hourly model output. The backward trajectories are terminated when they reach the thermal tropopause. The elapsed time is then the residual circulation transit time. A detailed description is given by Garny et al. (2014).

For the CLaMS simulation the RCTTs are calculated likewise by running backward trajectories, but in isentropic coordinates using the mean diabatic residual circulation velocities $(\overline{v^*}, \overline{Q^*})$, until they cross the 340 K surface in the tropics (between 30°N and 30°S). For the isentropic formulation in CLaMS the residual circulation velocities $(\overline{v^*}, \overline{Q^*})$ are calculated as the mass-weighted meridional and vertical wind velocities, based on the cross-isentropic vertical velocity $Q = \dot{\theta}$ (Andrews et al., 1987).

The different calculation frameworks for EMAC and CLaMS data (kinematic vs. diabatic vertical velocities) causes differences in the results, with a noisier structure for the kinematic vertical velocity (see Hoppe et al., 2016). However, as the internal vertical coordinates in EMAC and CLaMS are pressure and potential temperature, respectively, calculating residual circulation and mixing diagnostics in the two different coordinate systems is more consistent with the respective model simulation. For comparison between the two models, we interpolate the CLaMS results to pressure levels. Moreover, for comparing the data

it must be considered, that there are differences in the reference surface (tropopause in EMAC vs. 340K in CLaMS), which likely causes a difference in RCTT of 40-60 days, as Q*=0.7-1 K/day in that region. Another important aspect to note is the different treatment of trajectories at the model top. In CLaMS the data top is lower and trajectories are not considered, if they reach the top. So there might be lower transit times (missing data) at high altitudes and the results may not be so reliable in regions poleward about 60°N or 60°S.

In EMAC the top-level is higher and trajectories are artificially kept at the model top and advected horizontally in the top layer until they travel to lower levels. Due to the high model top at 0.01 hPa and weak vertical velocities there, the error for RCTTs calculated up to 5 hPa is small.

## 3.3 Calculation of aging by mixing on resolved and unresolved scales

Besides the transport through the residual circulation, AoA is a function of mixing (Neu and Plumb, 1999; Garny et al., 2014;

Ploeger et al., 2015b, a). As pointed out by Garny et al. (2014) mixing between the tropics and extratropics can cause additional aging by recirculation of aged air, which is mixed from the midlatitudes to the tropical pipe. This process is called "aging by mixing". In their study Garny et al. (2014) proposed that in global models aging by mixing can be interpreted as difference between simulated AoA and RCTT, assuming that mixing processes on unresolved scales (namely parameterized and numerical





diffusion) are small.

Recently, Ploeger et al. (2015b) calculated aging by mixing explicitly on resolved scales (in the following termed as "resolved aging by mixing") using the zonal mean isentropic tracer continuity equation (e.g. Andrews et al., 1987), which can be reformulated for AoA (e.g. Plumb, 2002). The formulation for the zonal mean continuity equation for AoA in isentropic coordinates

is explained in detail by Ploeger et al. (2015b) and by Ploeger et al. (2015a). For the CLaMS simulation, where the potential temperature is the vertical coordinate, this analysis is used to calculate the local mixing tendency ($\mathcal{M}$). Resolved aging by mixing is then given by integrating the explicitly calculated local mixing tendency $\mathcal{M}$ along a residual circulation trajectory ending at a given location and time, which is the path followed by this air parcel if advected by the residual circulation (Ploeger et al., 2015b).

As EMAC data are given on pressure coordinates the calculation of the local mixing tendencies must be adapted. The TEM continuity equation for zonal mean tracer concentrations in pressure coordinates is described by Abalos et al. (2013). Following Ploeger et al. (2015b) this equation can be used to derive the tendency equation for AoA ($\Gamma$). In the following the notation of Andrews et al. (1987) is used with overbars for zonal means and primes for the deviations from the zonal means. The AoA tendency equation (consisting of two residual circulation contributions and two eddy mixing contributions (for details see Ploeger

et al., 2015b) is given by:

$$\frac{\partial \Gamma}{\partial t} = 1 - \overline{v^*}\frac{\partial \Gamma}{\partial y} + e^{\frac{z}{H}}\frac{1}{cos(\varphi)}\frac{\partial M_y cos(\varphi)}{\partial y} - \overline{w^*}\frac{\partial \Gamma}{\partial z} + e^{\frac{z}{H}}\frac{\partial M_z}{\partial z} \tag{1}$$

Here, v* and w* denote the meridional and vertical component of the residual circulation respectively, z is the height, $H$ is the scale height (7km) and $\varphi$ denotes the latitude. The total local mixing tendency $\mathcal{M}$ is the sum of the horizontal and the vertical component of the local mixing tendency in the tendency equation for AoA (third and fifth term on the right side). The eddy

flux components $M_y$ and $M_z$ are defined as:

$$M_y = -e^{\frac{-z}{H}}(\overline{v'\Gamma'} - \frac{\overline{v'T'}}{S}\frac{\partial \Gamma}{\partial z}) \tag{2}$$

and

$$M_z = -e^{\frac{-z}{H}}(\overline{w'\Gamma'} - \frac{\overline{v'T'}}{S}\frac{\partial \Gamma}{\partial y}) \tag{3}$$

where $v'$, $w'$, $T'$ are the deviations of vertical and horizontal velocity and temperature from their zonal mean values. S denotes

a stability term defined by $S = H * N^2/R$ (with $H = 7km$, $R = 287m^2s^{-1}K^{-1}$ and $N^2$ being the Brunt-Väisälä frequency).

As mentioned above, resolved aging by mixing can be defined as the non-local, integrated mixing effect. This means integrating $\mathcal{M} = e^{\frac{z}{H}}\frac{1}{cos(\varphi)}\frac{\partial M_y cos(\varphi)}{\partial y} + e^{\frac{z}{H}}\frac{\partial M_z}{\partial z}$ along a residual circulation trajectory, gives the value of resolved aging by mixing at the starting location and time of the trajectory (Ploeger et al., 2015b):

$$\overline{\Gamma}(t) = RCTT + \int_{t_0}^{t} \mathcal{M}dt'. \tag{4}$$





As mentioned above, the effect of aging by mixing on mean age can be also obtained by building the difference between AoA and RCTT (Garny et al., 2014). This estimate is easier to deduce than the exact calculation (Eq. 1), but may include effects due to unresolved processes (see Garny et al., 2014). Understanding these unresolved processes may be important to explain

inter-model spread in AoA. Another advantage of calculating resolved aging by mixing is that the local mixing tendencies are available. Investigation of the local mixing tendencies provides a further insight into the processes causing AoA changes (Ploeger et al., 2015b). Subtracting "resolved aging by mixing" from "aging by mixing" provides the effect of mixing on unresolved scales, which is we define as "aging by diffusion".

## 4   The effect of residual transport, mixing and diffusion on AoA

### 4.1   Climatology of AoA, residual transport, resolved aging by mixing and aging by diffusion

The zonal annual mean of AoA, RCTT, resolved aging by mixing and aging by diffusion, averaged over the time period 1990-2011, for the simulations EMAC-RC1 (left column), EMAC-RC1SD (middle column) and CLaMS-ERAI (right coloum) is illustrated in Fig. 1. Note that the model data of CLAMS-ERAI are interpolated to pressure coordinates. All simulations show the typical pattern of AoA distribution with lower AoA in the tropical lower stratosphere and older air in the extratropical

middle stratosphere (see Fig. 1a). Comparing AoA of these simulations to observations shows that in EMAC-RC1 and in EMAC-RC1SD AoA is (a bit) younger at 50 hPa compared to MIPAS observations (see Jöckel et al., 2016, their figure 24). Note however that $SF_6$-derived AoA from MIPAS is larger compared to in-situ measurements (see e.g. Ploeger et al., 2015b), and that in CLAMS-ERAI AoA is in good agreement with observations in the lower stratosphere (see Ploeger et al., 2015b).

Before discussing the differences in the three model simulations, we investigate the effects that drive the AoA patterns. As shown in previous studies (e.g. Garny et al., 2014), AoA (Fig. 1a) largely differs from RCTT (Fig. 1b) in magnitude and structure for all simulations: RCTT is following the structure of the residual circulation. In most regions RCTT is lower than AoA, only at high latitudes in the lowermost stratosphere RCTT is higher. This shows that aging by mixing plays an important role for AoA. However as said before, parametrized and/or numerical diffusion is included in the aging by mixing term

(see section 3.3). Therefore we show resolved aging by mixing in Fig. 1. Consistently for all simulations resolved mixing leads to additional aging in most parts of the stratosphere, with maximum resolved aging by mixing in the midlatitude middle stratosphere, as mixing between the tropics and the extratropics leads to recirculation of air parcels. Only in the extratropical lowermost stratosphere, where mixing with tropospheric air occurs, resolved aging by mixing leads to a decrease in AoA. If looking at the pattern of the local mixing tendencies (see Fig. 2), a further insight into the processes causing the resolved

aging by mixing pattern is provided (Ploeger et al., 2015b). Large positive local mixing tendencies are present in the tropics and subtropics (in-mixing of aged air from high latitudes) and negative local mixing tendencies can be found at high latitudes for all three simulations. In the subtropics strongest positive local mixing tendencies occur below 50 hPa. This local mixing tendencies affect AoA above that level, so that resolved aging by mixing (Fig. 1c) increases with height, as more mixing levels





**Figure 1.** Zonal annual mean of (a) AoA, (b) RCTT, (c) resolved aging by mixing and (d) aging by diffusion from the years 1990-2011 for the simulations EMAC-RC1 (left), EMAC-RC1SD (middle), and CLAMS-ERAI (right). Units are given in years [a].

contribute to resolved aging by mixing (Garny et al., 2014; Ploeger et al., 2015b).

The effect of aging by diffusion (Fig. 1d) is showing that diffusion mainly leads to additional aging in all simulations. However, in general, the effect of aging by diffusion on AoA is relatively small (about 10%). The result, that diffusion mainly makes air older is interesting, because it was suggested before that diffusion is leading to too young age in models, because they





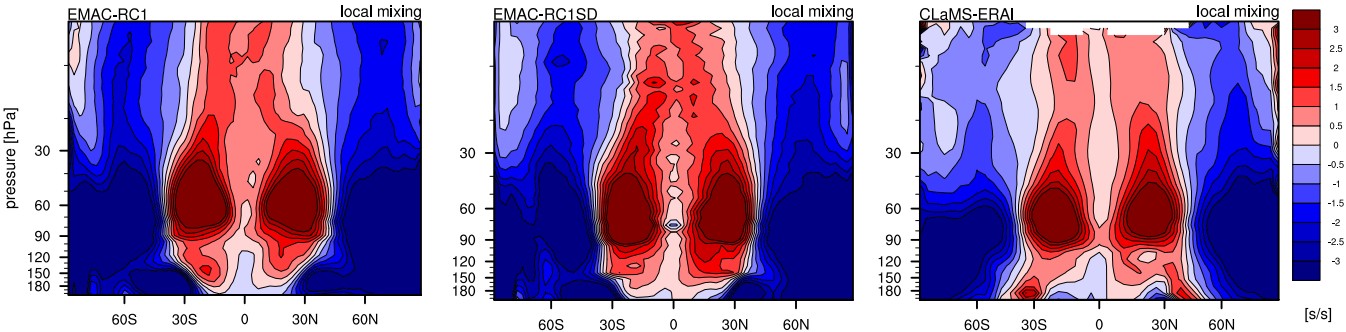

**Figure 2.** Zonal annual mean of the local mixing tendency (sum of horizontal and vertical contribution) on the AoA tendency budget (see Eq. 1). Average over 1990-2011. For EMAC-RC1 (left), EMAC-RC1SD (middle), and CLAMS-ERAI (right). Units are [s/s].

are too diffusive (e.g. SPARC CCMVal, 2010). For the two EMAC simulations the maximal values of aging by diffusion are found in southern high latitudes (at 30-60 hPa), while aging by diffusion is negative in mid-latitudes. In EMAC, unresolved diffusion is caused by numerical diffusion of the advection scheme and parameterized vertical diffusion. Assuming the former dominates, we expect strong local diffusion where AoA gradients are strong, and a strong local diffusion tendency, where the

second derivative of AoA is large (after Fick's law diffusion is proportional to $d^2 AoA/dy^2$). Aging by diffusion as shown in Fig. 1d is then the integrated effect over the local diffusion tendencies. The strong positive aging by diffusion effect at 60thus arises from the increasing gradient in AoA associated with the polar vortex. In CLAMS-ERAI, aging by diffusion is overall smaller compared to EMAC, and its pattern structurally strongly differs from EMAC. In CLAMS, unresolved diffusion arises from local subgrid scale mixing, that is flow-dependent and thus simulated in a physical manner.

Furthermore we analyze the differences in AoA (and in the effects that drive AoA) between the different model simulations. Although we have seen that the overall climatological structure agrees quite well for the shown simulations, there are differences in detail (see Fig. 1). Thus, in order to better compare the climatological structure, Fig. 3 presents the absolute differences between EMAC-RC1 and EMAC-RC1SD (left column) and between EMAC-RC1SD and CLAMS-ERAI (right





column) for AoA, RCTT, resolved aging by mixing, aging by diffusion and additionally for the local mixing tendencies. Note that for building the difference between EMAC and CLaMS we calculate the values relative to the same tropical reference layer of 100 hPa (not done in Fig. 1), as for CLaMS the transit time ends when the backward trajectories are crossing the 340 K surface, whereas for EMAC transit times are ending when trajectories cross the tropopause.

The difference in the free running simulation (EMAC-RC1) and the nudged simulation (EMAC-RC1SD) are presented in Fig. 3 (left column). EMAC-RC1 has somewhat lower AoA (up to 0.75 years) than EMAC-RC1SD in all regions, with largest differences occurring in the southern hemisphere. The younger air in EMAC-RC1 can be explained through lower residual circulation transit times (see Fig. 3b) due to faster circulation. It is known that the free running model overestimates plane-

tary wave activity in the southern hemisphere, that drives a stronger residual circulation and leads to a too weak polar vortex (Righi et al. (2015); Jöckel et al. (2016); Deckert and Cai, personal communication). Also quite big differences are present in the respective resolved aging by mixing pattern: in the tropical lower stratosphere and in polar regions less resolved aging by mixing, and at about 60°N and 60°S more resolved aging by mixing (being more pronounced in the southern hemisphere) is found in EMAC-RC1. These differences can be explained due to the fact that local mixing tendencies are weaker in the tropical

lower stratosphere in EMAC-RC1 compared to EMAC-RC1SD (consistent to the fact that the jet regions are less pronounced in EMAC-RC1 (see Righi et al., 2015; Jöckel et al., 2016)) and also in the polar regions. Local mixing tendency is stronger in the extratropical middle stratosphere (Fig. 3e), as residual circulation trajectories sample regions of negative or positive local mixing tendencies. Thus, besides the lower RCTT, an overall reduced resolved aging by mixing is also leading to the younger air in EMAC-RC1. Differences in aging by diffusion also can be found in the two EMAC simulations (Fig. 3d), showing the

opposite effect of resolved aging by mixing (Fig. 3c). As in EMAC-RC1 the polar jet (polar vortex) in the southern hemisphere is significantly too weak (see Righi et al., 2015; Jöckel et al., 2016), more mixing occurs cross the vortex edge, leading to a weaker gradient in AoA, and thus to less aging by diffusion.

Finally, we focus on the differences between EMAC-RC1SD and CLAMS-ERAI in Fig. 3 (right column). Mean AoA simu-

lated in EMAC-RC1SD and CLAMS-ERAI may differ by more than a year, despite the fact that the two simulations are both driven by ERA-Interim data. However, while CLaMS is directly driven by ERA-Interim data using diabatic heating rates as vertical velocities, the EMAC-RC1SD simulation is nudged to ERA-Interim horizontal winds and temperatures. As recently pointed out by Abalos et al. (2015), the residual circulation calculated from reanalysis data using different estimates (from the direct TEM residual velocities, from the momentum balance and from the thermodynamic balance, i.e. the diabatic circulation)

differs strongly in mean magnitude (up to 40%), likely due to data assimilation. The fact, that direct and momentum-based estimate are different in ERA-Interim is confirmed in Fig. 4a, where the vertical structure of the annual mean tropical up-welling over the 30°S-30°N latitude band, calculated with the direct estimate (black line) and the momentum-based estimate (blue line) is shown for ERAI-Interim (dashed lines) and also for the nudged simulation EMAC-RC1SD (solid lines). Note, that the estimates are plotted relative to the direct estimate of EMAC-RC1SD. The residual circulation in the nudged EMAC-

RC1SD simulation (black solid line) lies in between the direct and momentum-based estimate of the residual circulation of





ERA-Interim (Fig. 4a). And it also differs from the diabatic circulation in ERA-Interim (see Abalos et al., 2015, their figure 6), which is used in CLaMS. It is also interesting to note, that for EMAC-RC1SD the two residual circulation estimates are also different. In contrast the free-running simulation EMAC-RC1 the two estimates are nearly identical (figure not shown), as EMAC-RC1 simulates consistent data. Thus it is clear, that CLaMS-ERAI and EMAC-RC1SD have a different circulation due

to the different estimates of the residual circulation (this is also apparent in the different transit times of EMAC-RC1SD and CLAMS-ERAI in Fig. 3b). Furthermore, the EMAC-RC1SD simulation is nudged only up to 5 hPa, thus the circulation above also differs from ERA-Interim (see Figure 4a).

We have seen in Fig. 3a (right column) that CLAMS-ERAI presents lower AoA in most of the stratosphere (up to 1.25 years), with maximal values in the southern hemisphere. Only in the northern latitudes lower stratosphere younger air (up to -0.75

years) is apparent. These AoA differences are associated with differences in resolved aging by mixing, RCTT and aging by diffusion, all playing a similarly important role. EMAC-RC1SD shows mainly higher transit times (meaning slower circulation) in the southern hemisphere and in the northern hemisphere midlatitude middle stratosphere, whereas lower transit times are present in the northern lower stratosphere, consistent with lower AoA, here. Note that differences in calculating RCTT exist, as mentioned in section 3.2, with data being not comparable at high latitudes. Also resolved aging by mixing differs largely

between EMAC-RC1SD and CLAMS-ERAI (see Fig. 3c, left column), with resolved aging by mixing being mainly higher in EMAC-RC1SD (with maximum values in the midlatitude middle stratosphere), consistent with larger RCTTs, as a slower circulation also leads to a slower recirculation, and thus to higher resolved aging by mixing. Only at the edges of the polar vortex, mainly in the southern hemisphere, resolved aging by mixing is lower in CLAMS-ERAI. The pattern of the negative local mixing tendency differences in the lower midlatitude stratosphere at 30°S-60°S (Fig. 3e) roughly shows the reason for

this negative difference there. However, keep in mind, that it is difficult to interpret resolved aging by mixing with local mixing tendencies, as it is also affected by the residual circulation (as integrated effect).

In addition, aging by diffusion has a strong effect on the AoA difference pattern (Fig. 3d, right column) with significantly smaller aging by diffusion in CLAMS-ERAI throughout the stratosphere, in particular in the high latitude middle stratosphere of both hemispheres maximum values (up to 1 year) can be found. Here the representation of the advection certainly affects the

strength of aging by diffusion, and explains the strong difference pattern. As mentioned before in CLAMS-ERAI small-scale mixing is parametrized by anisotropic diffusion, simulated in a physical manner, being flow dependent. In contrast EMAC-RC1SD uses the flux-form semi-Lagrangian transport scheme, where (numerical) diffusion is more pronounced in regions of strong barriers (e.g. at the arctic and antarctic polar vortex), so unresolved diffusion is higher as in CLaMS there. This is consistent with Hoppe et al. (2014), who found in free-running simulations with the coupled modelsystem EMAC-CLaMS,

that transport barriers (polar vortex and tropical pipe) are stronger, if using the CLaMS tracer transport scheme compared to FFSL-transport scheme. The stronger transport barrier in CLaMS can be also seen in Fig. 1a, as the AoA gradient is stronger in CLAMS-ERAI. So local mixing across the transport barriers is less efficient (see also Fig. 3e), and thus less air is mixed to the tropical pipe and to polar regions. Correspondingly lower resolved aging by mixing and aging by diffusion is found in CLaMS.





## 4.2 Sensitivity: Role of enhanced subgrid scale mixing in ClaMS

Sensitivity studies were performed with CLaMS to test the sensitivity to parametrized subgrid scale mixing. In a sensitivity simulation (CLAMS-L1.0) the subgrid scale mixing strength was enhanced (critical Lyapunov exponent $\lambda_c = 1$ day$^{-1}$) as compared to the reference simulation CLAMS-L1.5 ($\lambda_c = 1.5$ day$^{-1}$). Choosing the smaller critical Lyapunov exponent of

$\lambda_c = 1$ day$^{-1}$ allows mixing to be triggered already at weaker flow deformations. Note that both choices of the small-scale mixing strength in CLaMS yield simulation results within the range of existing stratospheric observations (Konopka et al., 2004). Figure 5 (a-c) shows the zonal annual mean of AoA, resolved aging by mixing and aging by diffusion for the CLaMS simulations CLAMS-1.5 (left column) and CLAMS-1.0 (middle column). Additionally the right column of Fig. 5 displays the differences between CLAMS-1.0 and CLAMS-1.5. Enhancing small-scale model mixing increases simulated mean age of air

by a few months in most parts of the stratosphere (see Fig. 5). While the residual circulation in both simulations is exactly the same (equal RCTT's), this increase is, as expected, related to an increase in both resolved aging by mixing (Fig. 5b) and aging by diffusion (Fig. 5c). Aging by diffusion increases due to enhanced small-scale mixing mainly at the edge of the tropical pipe, where steep age gradients exist, and along the subtropical jets, where strong flow deformations frequently occur.

The diagnosed small-scale mixing intensity diagnosed from CLaMS (estimated as the percentage of grid points influenced

by parametrized small-scale mixing) consistently increases in these regions in the enhanced small-scale mixing simulation (see Fig. 6). Remarkably, enhanced small-scale mixing increases not only small-scale diffusion but also aging by mixing (Fig. 5b), even though the flow is exactly the same. This can be understood by the fact that enhanced AoA (by unresolved diffusion) automatically leads to enhanced resolved aging by mixing (the same mixing event leads to a larger exchange in AoA, or in other words the local mixing tendency, that is given by $\overline{v'AoA'}$ (see Eq. 2) increases, because AoA enhances). Therefore,

subgrid diffusion has a larger impact on AoA as diagnosed by aging by diffusion due to this feedback on mixing on resolved scales.

## 4.3 Mixing efficiency derived from the Tropical Leakly Pipe Model

Using the formulation of the conceptual Tropical Leakly Pipe model (Neu and Plumb, 1999), the so-called mixing efficiency can be defined as measure of the relative strength of mixing (for details see Garny et al., 2014). The mixing efficiency is defined

as the ratio of the mixing mass flux to the net mass flux across the tropical barrier. The mixing efficiency is proportional to the relative enhancement of AoA by mixing, and proved to be a useful measure of the relative mixing effects. Table 2 gives the mixing efficiency for the simulations discussed here. In the two EMAC simulations, the mixing efficiency is similar (0.43 and 0.44) despite the different underlying dynamics. The mixing efficiency is lower by about 10% in the CLAMS-ERAI simulation (0.39). In all simulations the mixing efficiency decreases when subtracting the effects of aging by diffusion, the difference

is on the order of 11% in EMAC and 7% in CLAMS-ERAI. As expected, in the CLAMS simulation with enhanced subgrid mixing (CLAMS-L1.0), the mixing efficiency is higher compared to the reference simulation CLAMS-ERAI. Overall, we find that unresolved diffusion enhances the mixing efficiency. This enhancement of the mixing efficiency can be explained by more diffusion across the tropical barrier, enhancing the two-way mixing mass flux, but not the net mass flux, so that the relative





**Table 2.** Mixing efficiency $\epsilon$ for the simulations EMAC-RC1, EMAC-RC1SD, CLAMS-ERAI, and CLAMS-L1.0. Mixing efficiency is derived with the TLP model, with a tropical pipe bounded by 30°N -30°S. The upper row gives the mixing efficiency $\epsilon$(AoA) using the full AoA values, the lower row the mixing efficiency $\epsilon$(RCTT+ aging by resolved mixing(Amix)).

|  | EMAC-RC1 | EMAC-RC1SD | CLAMS-ERAI | CLAMS-L1.0 |
|---|---|---|---|---|
| $\epsilon$(AOA) | 0.44 | 0.43 | 0.39 | 0.41 |
| $\epsilon$(RCTT+Amix) | 0.39 | 0.38 | 0.36 | 0.37 |

mixing strength increases. The effects of unresolved mixing on the mixing efficiency are on the order of 10%. Consistent with stronger aging by diffusion in EMAC compared to CLaMS, the mixing efficiency is higher in EMAC. The mixing efficiency appears to be a useful measure of relative mixing strength, that can be considered a model property that is affected by the numerics in the model advection (and other relevant parametrizations), rather than by the underlying dynamics (as seen by the almost identical mixing efficiency in the two EMAC simulations).

### 4.4 Trend of AoA, residual transport, resolved aging by mixing and aging by diffusion

Figure 7a presents the zonal annual mean trend of AoA from 1990 to 2011, calculated as linear trend. Strippling shows regions where trends are not significantly different from zero at the 95% level. To understand the processes that contribute to AoA changes, also the zonal annual mean trends of RCTT, resolved aging by mixing and aging by diffusion are shown in Fig. 7b-d. Again the trends for the simulations EMAC-RC1 (left column), EMAC-RC1SD (middle column) and CLAMS-ERAI (right column) are given in this panel plot. The AoA trend (Fig. 7a) shows a significant decrease throughout the stratosphere in all simulations. Only in CLAMS-ERAI a small positive trend is apparent in the middle stratosphere at 30°N-60°N. The negative AoA trend is in good agreement with other CCM simulations (see e.g Butchart et al., 2010). Compared to the balloon-borne in-situ AoA measurements of Engel et al. (2009), which cover the period 1975-2005, only the CLAMS-ERAI simulation confirms their observed, insignificant slightly positive trend in the northern hemisphere subtropics and midlatidues above about 30 hPa. Furthermore, it has been shown by Ploeger et al. (2015a) that the AoA trend (2002-2012) in CLAMS-ERAI also agrees well to the observed AoA trend of the MIPAS satellite instrument. In contrast, the two EMAC simulations, also the one nudged to ERA-Interim, do not reproduce the observed trend patterns. However, as discussed in section 4.1 the residual circulation in reanalysis data suffer from large inaccuracies, so it is not surprisingly, that CLaMS-ERAI and EMAC-RC1SD have different trends in AoA and RCTT. Again a closer look at the vertical structure of the tropical upwelling trend (see Fig.4b, trends are plotted relative to the climatological direct tropical upwelling) shows, that in EMAC-RC1SD the trend of direct tropical upwelling (black solid line) is not identical to the momentum-based estimate of tropical upwelling (blue solid line), below 5 hPa, where nudging is applied. The comparison to ERA-Interim (dashed lines) shows, that the trend in direct tropical upwelling is completely different in EMAC-RC1SD and ERA-Interim, the same is the case for the momentum-based estimate. Furthermore, the EMAC-RC1SD simulation is nudged only up to 5 hPa, thus the circulation above also is not constrained by ERA-Interim.





It is important to understand the processes that drive the AoA trends. In all three simulations, trends in resolved aging by mixing (Fig. 7c) contribute more to the overall AoA trend than trends in RCTT (Fig. 7b). However, trends in resolved aging by mixing result not only from trends in local mixing tendencies, but also from changes in the residual circulation, as changes

in RCTT change the time exposed to mixing (see Garny et al., 2014; Ploeger et al., 2015a). Those effects can be separated by calculating resolved aging by mixing with fixed local mixing tendencies (see Ploeger et al., 2015a). Figure 8 summarizes the zonal annual mean resolved aging by mixing trend due to circulation change for the simulations EMAC-RC1 (left column), EMAC-RC1SD (middle column) and CLAMS-ERAI (right column). All simulations agree that the resolved aging by mixing trend due to residual circulation change alone can explain the resolved aging by mixing trends above about 30 hPa (Fig. 8).

This is consistent with Ploeger et al. (2015b), who found the strongest effect is above about 550K. Below, local mixing is relevant for the resolved aging by mixing trend. In contrast, the effect of aging by diffusion on the AoA trend is very small with large regions being not significant in the EMAC simulations (see Fig. 7d). However, in CLaMS the trend of aging by diffusion significantly impacts AoA. Note however, that the aging by diffusion trend pattern in CLaMS is influenced by the CLaMS RCTT calculation, which is not reliable in regions poleward about 60°N or 60°S, as many residual circulation trajectories get

lost in these regions (see section 3.2).

Figure 7 also reveals the differences in the trend patterns between the model simulations. We begin with comparing the trend pattern of EMAC-RC1 and EMAC-RC1SD (Fig. 7, left and middle column). Both simulations have a negative trend throughout the stratosphere, with strongest trend in the northern hemisphere middle stratosphere above 40 hPa. However, the AoA trend

in EMAC-RC1 is notably weaker with largest differences in the southern hemisphere polar region. To explain these differences we have a closer look at the differences in the trends of the RCTT, aging by mixing and aging by diffusion. The most important differences are given by the aging by mixing trends, with a weaker trend in EMAC-RC1. The increase in the residual circulation is stronger in the RC1SD simulation, as seen by the stronger trends in RCTT. However, also the trends in RCTT impact the differences in the AoA trend: There is a much stronger trend in the northern hemisphere at about 60°N in EMAC-RC1SD,

which might be related to the too weak vortex. Differences in the trends of aging by diffusion are difficult to interpret as they are mainly not significant.

Finally we have a closer look at the notable differences between the AoA trend in EMAC-RC1SD and CLAMS-ERAI (Fig. 7, middle and right column). Although AoA decreases in most of the stratosphere in both simulations, CLAMS-ERAI shows

the highest negative trend in the southern stratosphere, EMAC-RC1SD in contrast, in the northern hemisphere. The slightly positive, but insignificant trend in the northern hemisphere at 30 hPa cannot be found in EMAC-RC1SD. In contrast, EMAC-RC1SD shows maxima in the negative AoA trends in that region. A closer inspection of the components, that drive these AoA trends reveals that both the trend in RCTT and the trend in resolved aging by mixing play a role (as mentioned before). Another important difference is found in the trend in aging by diffusion, which has a significant effect on the AoA trend in

CLAMS-ERAI (although not explaining the increase in AoA in the northern hemisphere). Particularly in the southern polar





vortex a large negative trend can be found, however as mentioned before the CLaMS aging by diffusion pattern in regions poleward about 60°N or 60°S is not so reliable. This is not the case in the EMAC-RC1SD simulation. This fact is consistent with the parametrized subgrid scale mixing in CLaMS, which is flow depended. So subgrid scale mixing is underlying a trend and as the southern hemisphere polar vortex is getting stronger in a cooling stratosphere (e.g. Thompson and Solomon, 2002),

aging by diffusion decreases there.

## 5 Summary and Conclusion

This study presents a comparison of the annual zonal mean AoA and of the AoA trends in three simulations using the two different models EMAC and CLaMS. To understand the AoA pattern we analyze the effects that drive AoA and AoA trends. These effects include residual circulation transit time, resolved aging by mixing and aging by diffusion. We calculate the resid-

ual circulation transit time (RCTT) and interpret the difference between AoA and RCTT as aging by mixing. However, as parametrized (e.g. vertical) diffusion or numerical diffusion are included in this difference, we further calculate resolved aging by mixing (by integrating the daily local mixing tendencies numerically along the residual circulation trajectories). By building the difference of aging by mixing and resolved aging by mixing, we introduce a method to determine aging by diffusion.

The effect of aging by diffusion on AoA has a considerable effect on AoA, mostly leading to additional aging in all simulations. This finding is confirmed by a CLaMS sensitivity calculation, where subgrid scale mixing was enhanced. Enhancing subgrid scale mixing leads to an increase in aging by diffusion, making air older. We further found that the spatial distribution and strength of resolved aging by mixing strongly depends on the type of advection scheme used in the model. EMAC, which has an advection scheme including numerical diffusion, shows larger AoA and mixing efficiency than CLaMS, where unre-

solved diffusion arises only from parametrized subgrid scale mixing that is flow dependent and thus more physical. Overall the effect of aging by diffusion on AoA and on the mixing efficiency is in the order of 10%. However, this is a lower estimate due to the dependence of unresolved mixing on resolved aging by mixing, that is not captured by our method. Consequently, at least some of the spread in AoA between different models is likely to be be caused by unresolved diffusion.

For the trends in AoA we found that they are largely driven by changes in resolved aging by mixing for all simulations, consistent with the study of Ploeger et al. (2015a). We further verified the result of Ploeger et al. (2015b), that the trends in resolved aging by mixing above 30 hPa are likely explained by changes in the residual circulation rather than in changes of local mixing tendencies. In the lower stratosphere also changes in local mixing are important. AoA trends of the two models show considerable differences, despite being nudged to the same dynamics. However, as recently discussed by Abalos et al.

(2015), the residual circulation calculated from reanalysis data differs strongly, if different estimates are used as it is the case in EMAC and CLaMS. Therefore, model simulations driven by or nudged by reanalysis have to be interpreted with care with respect to circulation and circulation changes. Furthermore, as the EMAC-RC1SD simulation is nudged only up to 5 hPa, the circulation above is not constrained by ERA-Interim. To quantify the effect of nudging on the residual circulation sensitivity





studies with varying nudging height could be performed. Regarding the effect of aging by diffusion on the AoA trend, we conclude, that unresolved diffusion has a minor mostly not significant impact on AoA trends in EMAC. However, in the CLaMS simulation we have a significant, small effect on the AoA trend. Comparing the simulated AoA trend to observations, for the EMAC simulations the discrepancy between observed and simulated AoA changes, still exists and cannot be explained by the

5  trend in aging by diffusion. Strongest differences between CLAMS-ERAI and EMAC-RC1SD are found in the resolved aging by mixing trend (positive in the northern hemisphere for CLaMS, and negative for EMAC), but differences could not be traced back to process level. In contrast, the CLaMS simulation is in agreement with observations (see also Ploeger et al., 2015b), providing support for the quality of the vertical velocities derived from ERA-Interim diabatic heating rates. Note here, that the observational trend results contain uncertainties due to limited spatial coverage, short data record and technical aspects

10  regarding the deviation of AoA from trace gases (see Garcia et al., 2011). Overall, the discrepancy between observed AoA trends, reanalysis-driven AoA estimates and AoA trends in models is not resolved yet.

*Author contributions.* Simone Dietmüller, Hella Garny and Felix Plöger made substantial contributions to conception and design, analysis and interpretation of the data. Moreover they participated in drafting the article. Patrick Jöckel and Duy Cai, set up and ran the EMAC model simulations and Flelix Plöger the CLaMS simulations.

15  *Acknowledgements.* This work was supported by the HGF Young Investigators Group MACClim (The middle atmosphere in a changing climate). The EMAC simulations were done within the project ESCiMo (Earth System Chemistry integrated Modelling), a national (German) contribution to the Chemistry Climate Model Initiative, and have been performed at the German Climate Computing Centre DKRZ through support from the Bundesministerium für Bildung und Forschung (BMBF). Felix Ploeger was supported by the HGF Young Investigators Group A-SPECi (Assessment of stratospheric Processes and their effects on climate variability).



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





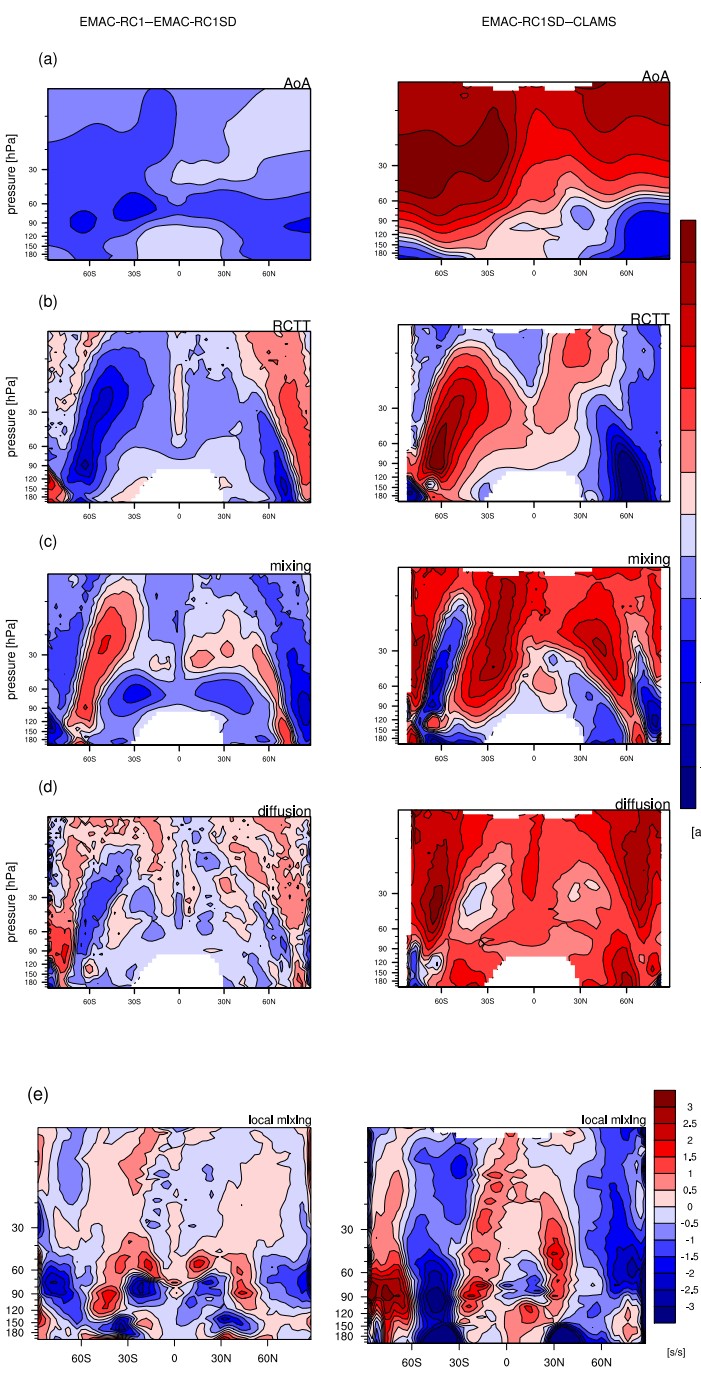

**Figure 3.** Zonal annual mean of the absolute differences in (a) AoA, (b) RCTT, (c) resolved aging by mixing, (d) aging by diffusion between EMAC-RC1 and EMAC-RC1SD (left), and between EMAC-RC1SD and CLAMS-ERAI (right). Moreover, the absolute differences for the local mixing tendencies are shown (e). Note that the difference between EMAC-RC1SD and CLAMS-ERAI is calculated relative to the same reference layer.





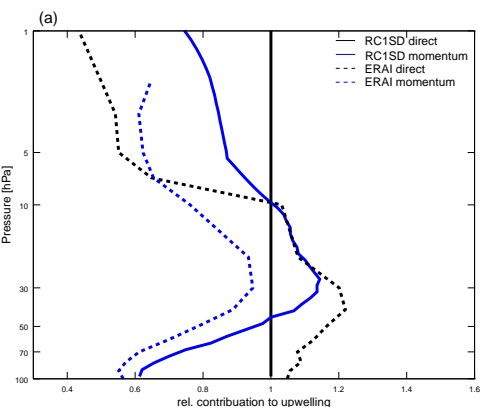
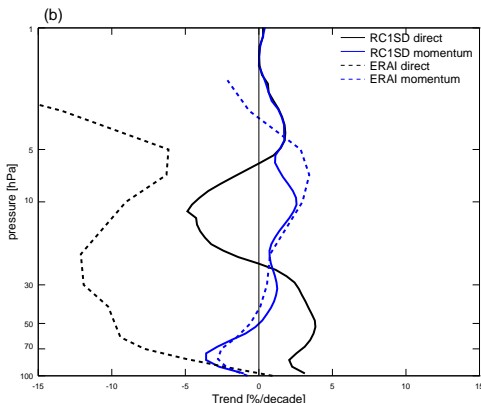

**Figure 4.** (a) Annual mean profiles of tropical upwelling ( 30°S-30°N), calculated with the direct (black) and momentum-based (blue) estimate for the simulations EMAC-RC1SD (solid) and ERA-Interim (dashed). Note that tropical upwelling is plotted as relative contribution to the total direct upwelling of EMAC-RC1SD. (b) Trends (1990-2011) for the profiles of tropical upwelling (relative to the climatological direct estimate of tropical upwelling in EMAC-RC1SD), again calculated with direct and momentum-based estimate and for EMAC-RC1 and ERA-Interim. The trend is given in [% per decade].





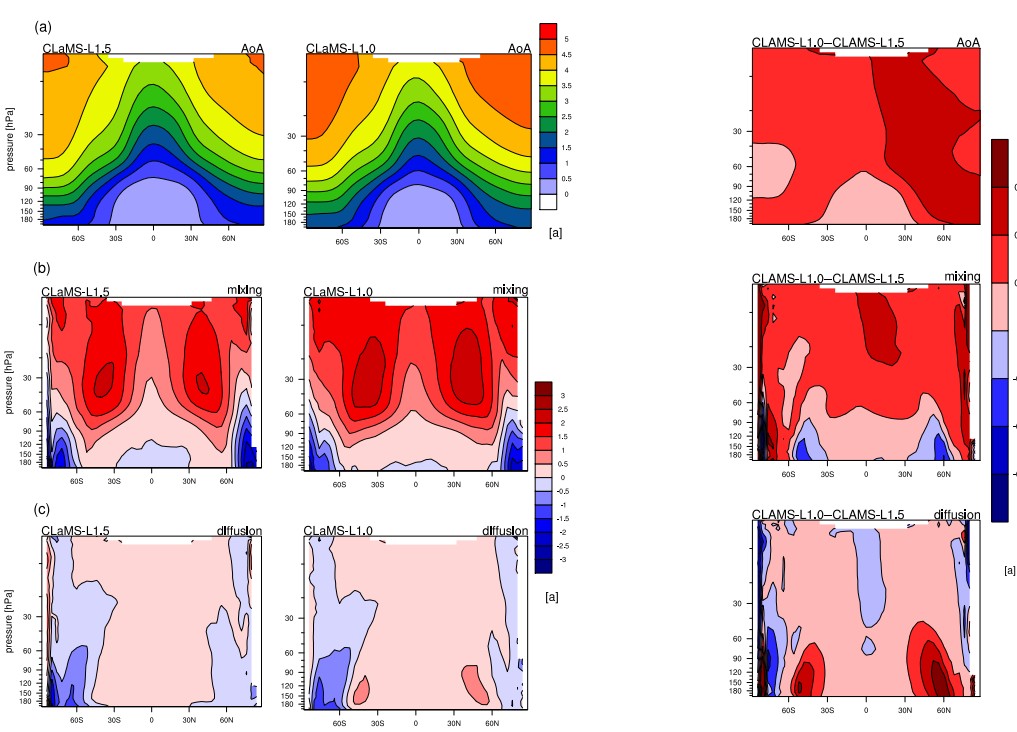

**Figure 5.** Sensitivity simulation with respect to mixing strength in CLaMS: Zonal annual mean of (a) AoA, (b) resolved aging by mixing and (c) aging by diffusion from the years 1990-2010 for the simulations CLAMS-1.5 (left) and CLAMS-1.0 (middle). Moreover, respective absolute differences between CLAMS-1.0 and CLAMS-1.5 are given at the right column. Units are given in years [a].





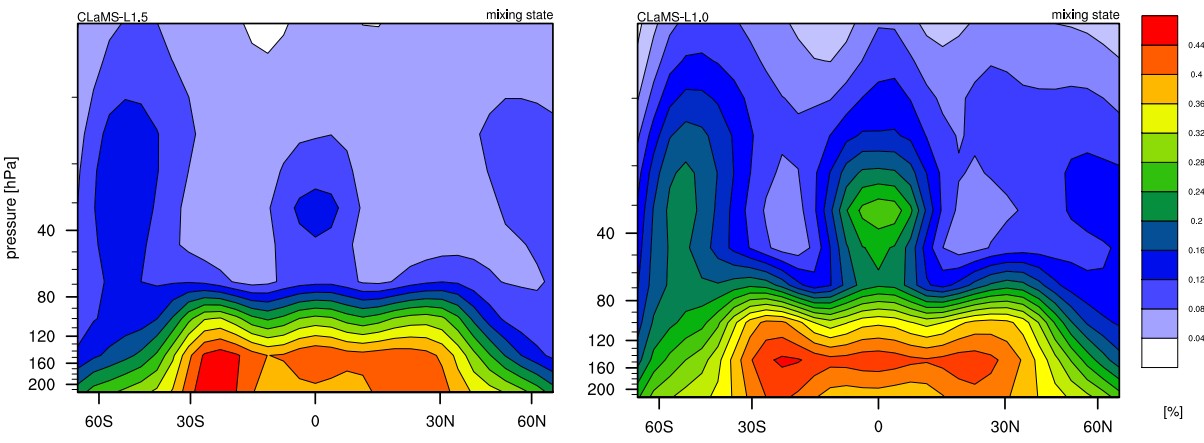

**Figure 6.** Diagnosed small-scale mixing intensity (estimated as the number of gridpoints influenced by the parametrized CLaMS mixing in %) for the simulations CLAMS-L1.5 (left) and CLAMS-L1.0 (right)





**Figure 7.** Trends for 1990-2011 of (a) AoA, (b) RCTT, (c) resolved aging by mixing and (d) aging by diffusion for the simulations EMAC-RC1 (left), EMAC-RC1SD (middle), and CLAMS-ERAI (right). Black contours show climatological values. Stripling displays regions where trends are not significant on a 95% level. Units are given in [year/decade].





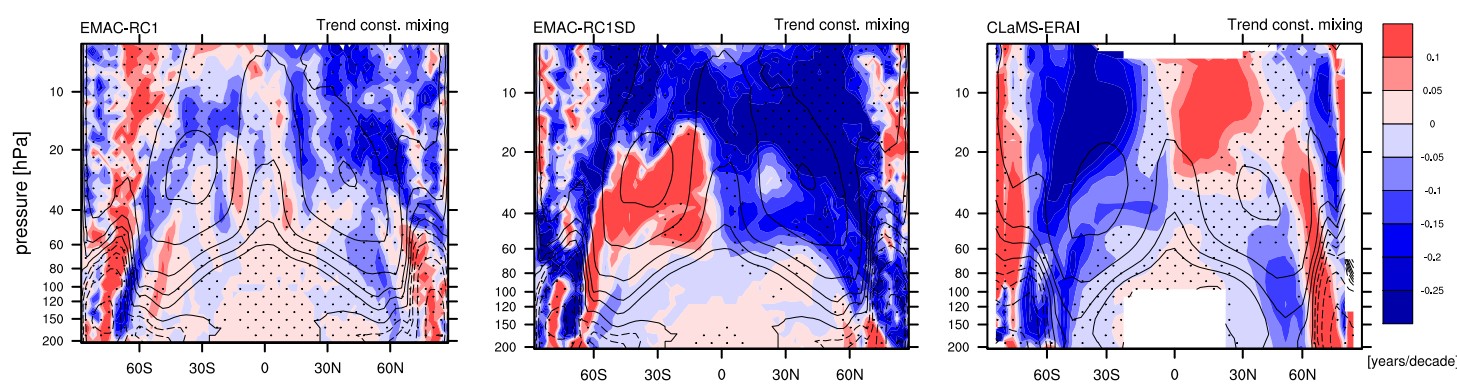

**Figure 8.** Trends (1990-2011) in resolved aging by mixing from residual circulation change alone for the simulation EMAC-RC1 (left), EMAC-RC1SD (middle) and CLAMS-ERAI (right). Stripling displays statistical significance on a 95% level. Units are given in [year/decade].