# Peer review of "Effects of mixing on resolved and unresolved scales on stratospheric age of air"

_Atmospheric Chemistry and Physics, 2016_

## Referee Comment (RC1) · Anonymous Referee #2 · 13 Feb 2017

ACP-2016-1144

Effects of mixing on resolved and unresolved scales on stratospheric age of air

General:

1. The paper is one of the first to quantitatively separate aging due to "unresolved" vs. "resolved" mixing throughout the stratosphere. A conclusion is made regarding how diffusion actually makes air slightly older (at least in these models), which contradicts some previous thoughts. This could be a significant text for others to use in the future, as we continue to look into why models differ in transport and trends, and how we can relate these results to observations.

2. Overall the paper is succinct and easy to read. My concerns are only medium/minor:

[Figure]

expanding on ideas (so reproduction by others will be better possible), making figures clearer, and fixing syntax/grammar issues.

Issues throughout the paper:

1. The residual circulation advection terms (v-bar-star and w-bar-star) should have the bar only over the v/w, not over the star.

2. Could you make all the figures have the same y-scale lengths, tickmarks, and tick-mark labels? Could you increase the font on the figures (including the colorbar values)? Also, how would Fig. 4 look if it went down to 200hPa like the other figures?

3. Could you say "vertical advection by the residual circulation" instead of "tropical up-welling"? I think some people confuse "tropical upwelling" with "total upward transport" which is the sum of advection and vertical diffusion. Or just make a note that "tropical upwelling" only refers to vertical advection term and nothing else.

Individual issues:

1. p 5, line 17: How would the results change if you varied this zonal band seasonally? Don't need to re-run experiments, just acknowledge.

2. p 6, line 4-5: Is any info lost by using monthly means (instead of daily)?

3. p 6, line 11-12: Explain why this is 30S-30N and not 10S-10N? How wide do you think the zonal band is for the EMAC calculation (that uses the thermal tropopause)? Is it similar?

4. p 6, line 19: Do you interpolate the data on every lat,lon gridpoint? Or do you interpolate zonal mean data?

5. p 6, line 22: remove comma: "…are not considered[,] if they…"

6. p. 7, line 18: Could you give a mean value of H in the areas you are looking? Does it equal 7km in every model?

7. p 7, line 23 (equation 3): Make sure prime (zonal deviation) is under the overbar (for v'T'). Also I think it should be a plus (instead of minus) inside the parenthesis. I imagine this is correct in your code, but double check.

8. p 8, line 4-5: Great job here.

9. p 10, line 3-4: You assume that numerical diffusion dominates in EMAC. Is this a good assumption? Can you cite anything?

10. p 10, line 6: Should this sentence "...by diffusion effect at 60thus arises..." instead say "...by [the] diffusion effect at 60S thus arises..."?

11. p 10, line 14: (talking about Fig. 3) I was wondering, do the (lapse rate) tropopauses (not the entry levels used for the backward trajectories) differ between any of the models? This would be helpful to know, because it could affect the difference plots (shifting the signals in height by a bit).

12. p 11, line 29: insert comma "...velocities, from the momentum balance[,] and from..."

13. p 11, line 20: remove comma "The fact[,] that direct..."

14. p 11, line 31: add letter "estimate[s] are different..."

15. p 11, line 33-34: really emphasize that the EMAC-RC1SD line = 1 everywhere. For example, add: "...relative to the direct estimate of EMAC-RC1SD [such that the SD (black solid line) equals one throughout the profile]." or something like that.

16. p 12, line 2-3: reword this sentence to be "It is interesting that the two residual estimates are also different for EMAC-RC1SD."

17. p 12, line 13: reword "...northern lower stratosphere, consistent with lower AoA there."

18. p 12, line 15: Should this say "right column"?

19. p. 12, line 18: reword, do you mean "higher" for CLAMS? Is this regarding the blue part (around 60S, 10-100hPa) of Fig. 3c (right column)?

20. p 12, line 28: reword, do you mean: is higher [than] in CLAMS?

21. p 13, line 9-10: Great work here.

22. p 13, line 14-15: How do you identify the gridpoints influenced by small-scale mixing? Is this quantity given by the model? Or is this just mathematically derived from the aging by diffusion value? How would another scientist find this?

23. p 14, line 4-5: I understand what you are saying, but could you briefly talk about how nudging might (or might not) counteract the underlying dynamics?

24. p 15, line 10: What latitude band? And what mean pressure level coincides with 550K?

25. p 15, line 25: reword "..might be related to a weaker vortex."

26. p 16, line 9: insert "...and [unresolved] aging by diffusion."

27. p 16, line 17: Can you stress how this finding differs from past work? Citations.

28. p 16, line 28: Which two models? Be specific here.

29. p. 16, line 33: insert comma: "...nudging on the residual circulation[,] sensitivity studies..."

30. p 17, line 1: insert s: "...varying nudging height[s] could..."

31. p 17, line 1-2: remove comma: "...on the AoA trend, we conclude[,] that unresolved..."

32. p 17, line 2: insert commas and reword "...has a minor[,] mostly non-significant[,] impact on AoA..."

33. p 17, line 3-5: very confusing to me, reword: "The AoA trend discrepancy between

**[ACPD](ACPD)**

Interactive
comment

observations and both EMAC simulations still exists but cannot be explained by the trend in aging by diffusion." ...or something like that.

34. p 17, line 6-7: What does this mean? Elaborate on "difference could not be traced back to process level." What is the process level?

35. p 17, line 8: remove comma: "Note here[,] that..."

36. p 17, line 9: insert comma: "...short data record[,] and technical..."

37: p 17, line 11: insert comma: "...AoA estimates[,] and AoA trends..."

Figure 8: Add text about the "black contours show climatological values".

---

## Referee Comment (RC2) · E. Ray (Referee) · 21 Feb 2017

This paper uses output from two global models to examine in detail the individual components that determine the modeled mean age of air. This analysis is unique and very interesting in that it shows specifically how different the mean age can be among the same model free running vs. nudged, and two different models using the same meteorology as input. The unresolved mixing contribution to the age of air is not large overall but still interesting to see it's contribution and how different it is among the models. This type of analysis can help explain why models differ among themselves and from observations.

My only criticism of the paper is the poor grammar in many places. I've tried to make comments on some obvious errors but the paper could use further editing. Overall the

techniques of the paper are well described and the figures are clear. I recommend publication with consideration of the minor grammatical comments below.

Specific comments:

Pg 1, line 22: add comma after "AoA", add "such" before "as" and remove "e.g."

Pg. 2, line 4: "...which used the output from 15 CCMs..."

Pg. 2, line 7: remove "if"

Pg. 2, line 13: "...(Stiller et al., 2012), which show..."

Pg. 3, line 27: add comma after "EMAC"

Pg. 5, line 20: 340 K is much below the tropical tropopause, why not use 380 K?

Pg. 6, line 3: "...simulation output, RCTTs are..."

Pg. 6, line 22: remove the comma

Pg. 6, line 24: "...poleward of about..."

Pg. 6, line 32: "...as the difference..."

Pg. 8, line 2: remove "building"

Pg. 8, line 8: remove "is"

Pg. 8, line 12: change "is" to "are"

Pg. 8, line 16: change "is" to "are"

Pg. 8, line 22 "...RCTT follows the..."

Pg. 8, line 32: change "this" to "these"

Pg. 9, line 2: change "is showing" to "shows"

Pg. 9, line 4: change "is leading" to "leads"

Pg. 10, line 6: "...60S thus..."

Pg. 11, line 4: change "are ending" to "end"

Pg. 11, line 15: change "to" to "with"

Pg. 11, line 16-18: Not sure what you are trying to say in this sentence.

Pg. 11, line 18: "...mixing also leads to..."

Pg. 11, line 30: remove comma after "fact"

Pg. 11, line 31: change "estimates"

Pg. 12, line 3: "In contrast, in the..."

Pg. 12, line 4: move comma from after "clear" to after "Thus"

Figure 4a: The label on the x-axis, "rel. contribution to upwelling", is a bit confusing. I assume what you're showing here is the magnitude of the upwelling relative to the direct RC1SD estimate expressed as a fraction. If that's the case, the label should read more like "Relative Magnitude Compared to RC1SD direct".

Pg. 12, line 8: change "presents" to "has"

Pg. 12, line 28: change "as" to "than"

Pg. 12, line 29: "model system" is two words

Pg. 13, line 5: remove "already"

Pg. 14, line 7: change to "Stippling"

Pg. 14, line 17: change "also" to "including"

Pg. 14, line 19: change to "surprising"

Pg. 15, line 35: change to "not explaining" to "does not explain"

Pg. 16, line 3: change "depended" to "dependent"

Pg. 17, line 10: Aspects of the uncertainties in the observed AoA trend that you mention were examined in Ray et al. (2015) so that should at least be included as a reference along with Garcia et al.

―――――――――――――――――

---

## Author Comment (AC1) · 24 Apr 2017

**Reply to Anonymous Referee #2 (ACP-2016-1144)**

We thank the reviewer for the positive and constructive comments on our manuscript. Below, we summarize our answers to all questions of Referee #2. Moreover the manuscript is changed taking into account the questions and comments (the changed manuscript, with changes highlighted, is attached to the reply).

1. The residual circulation advection terms (v-bar-star and w-bar-star) should have the bar only over the v/w, not over the star.

 $\rightarrow$  corrected

2. Could you make all the figures have the same y-scale lengths, tickmarks, and tickmark labels? Could you increase the font on the figures (including the colorbar values)? Also, how would Fig. 4 look if it went down to 200hPa like the other figures?

→ Figures are changed accordingly. However the y-length of Fig. 4 was not changed down to 200hPa, as the **tropical** mean profile of wstar is shown (tropical tropopause lies at about 100 hPa).

3. Could you say "vertical advection by the residual circulation" instead of "tropical upwelling"? I think some people confuse "tropical upwelling" with "total upward transport" which is the sum of advection and vertical diffusion. Or just make a note that "tropical upwelling" only refers to vertical advection term and nothing else.

→ Tropical upwelling was not renamed in the manuscript, as is it is mainly used to describe the upwelling through the residual circulation, however it is mentioned now, that tropical upwelling is the vertical advection along the residual circulation (see p.12, line 1).

**Individual issues:**

1. p 5, line 17: How would the results change if you varied this zonal band seasonally? Don't need to re-run experiments, just acknowledge.

→ AoA results should not change much, if the zonal band is varied seasonally. Looking at the zonal mean AoA in Figure 1a, shows that the AoA gradient is very flat at 100hPa  $30^{\circ}S-30^{\circ}N$ . Acknowledged in the manuscript.

2. p 6, line 4-5: Is any info lost by using monthly means (instead of daily)?  $\rightarrow$  The results of RCTTs differ only little whether daily or monthly values of the residual circulation are used.

3. p 6, line 11-12: Explain why this is 30S-30N and not 10S-10N? How wide do you think the zonal band is for the EMAC calculation (that uses the thermal tropopause)? Is it similar?

→ The latitude band  $30^{\circ}S-30^{\circ}N$  is used, as the latitude band  $10^{\circ}S-10^{\circ}N$  would be to narrow (trajectories coming down poleward of  $10^{\circ}S-10^{\circ}N$  would then be lost).

We did not explicitly calculate the zonal band in EMAC. However, Birner and Bönisch 2011 show in their Fig. 5 the annual mean values of the entry latitudes: they found that for the deep circulation branch trajectories enter at 5° and for the shallow branch close to the poleward flanks of the tropics. This should be similar in EMAC.

4. p 6, line 19: Do you interpolate the data on every lat,lon gridpoint? Or do you interpolate zonal mean data?

 $\rightarrow$  We interpolate zonal mean output of ClaMS. (Now explicitly mentioned in the manuscript)

5. p 6, line 22: remove comma: "are not considered[,] if they"  $\rightarrow$  done

6. p. 7, line 18: Could you give a mean value of H in the areas you are looking? Does it equal 7km in every model?

→ We use 7 km for H, as it is the standard assumption. H varies between 6.4km-7km in the stratosphere. However H should not vary much between the models, as temperatures are not too different between the model simulations. As EMAC-RC1SD and CLaMS are both driven by ERA-Interim, temperatures are very similar.

7. p 7, line 23 (equation 3): Make sure prime (zonal deviation) is under the overbar (for v'T'). Also I think it should be a plus (instead of minus) inside the parenthesis. I imagine this is correct in your code, but double check.

 $\rightarrow$  You are right! In the code it was done correctly.

8. p 8, line 4-5: Great job here.

9. p 10, line 3-4: You assume that numerical diffusion dominates in EMAC. Is this a good assumption? Can you cite anything?

→ Unfortunately there are no citations. It is a conceptional idea: if vertical diffusion makes AoA younger (not tested within EMAC, as we have no separation of vertical and horizontal diffusion; but we could show in the TLP model, that vertical diffusion leads to a decrease in tropical AoA), then numerical diffusion should dominate (because aging by diffusion makes air older). Manuscript adopted here. 10. p 10, line 6: Should this sentence "by diffusion effect at 60thus arises" instead say "by [the] diffusion effect at 60S thus arises"?

→ yes.

11. p 10, line 14: (talking about Fig. 3) I was wondering, do the (lapse rate) tropopauses (not the entry levels used for the backward trajectories) differ between any of the models? This would be helpful to know, because it could affect the difference plots (shifting the signals in height by a bit).

→ We did not quantify the model differences in the tropopause height, but we considered the fact, that the tropopause is different in the models: thus we are building the difference in Figure 3 relative to 100 hPa (however this was noted in the text, see p.11, line 2-4, old manuscript).

12. p 11, line 29: insert comma "...velocities, from the momentum balance[,] and from...."  $\rightarrow$  done

13. p 11, line 20: remove comma "The fact[,] that direct...."  $\rightarrow$  done

14. p 11, line 31: add letter "estimate[s] are different...."  $\rightarrow$  done

15. p 11, line 33-34: really emphasize that the EMAC-RC1SD line = 1 everywhere. For example, add: "relative to the direct estimate of EMAC-RC1SD [such that the SD(black solid line) equals one throughout the profile]." or something like that.  $\rightarrow$  done

16. p 12, line 2-3: reword this sentence to be "It is interesting that the two residual estimates are also different for EMAC-RC1SD."  $\rightarrow$  done

17. p 12, line 13: reword "northern lower stratosphere, consistent with lower AoA there."  $\rightarrow$  *done*

18. p 12, line 15: Should this say "right column"? → Yes, you are right.

19. p. 12, line 18: reword, do you mean "higher" for CLAMS? Is this regarding the blue part (around 60S, 10-100hPa) of Fig. 3c (right column)?

 $\rightarrow$  Yes, you are right.

20. p 12, line 28: reword, do you mean: is higher [than] in CLAMS?  $\rightarrow$  Yes.

21. p 13, line 9-10: Great work here.

22. p 13, line 14-15: How do you identify the gridpoints influenced by small-scale mixing? Is this quantity given by the model? Or is this just mathematically derived from the aging by diffusion value? How would another scientist find

→ To diagnose the intensity of small-scale mixing in CLaMS, CLaMS air parcels affected by mixing (merging or insertion) were flagged, such that for the zonal mean plot of Fig. 6 the percentage of these affected air parcels in each lat/level grid box could be calculated. Now clarified in the text.

23. p 14, line 4-5: I understand what you are saying, but could you briefly talk about how nudging might (or might not) counteract the underlying dynamics?

 $\rightarrow$  In the simulation without nudging, enhanced wave forcing is consistent with stronger mixing and stronger circulation. This is not the case in the nudged simulation, here inconsistent wave forcing influences the underlying dynamics. However as mentioned in the text the underlying dynamics has not a big influence on the mixing efficiency, see table 2.

24. p 15, line 10: What latitude band? And what mean pressure level coincides with 550K?

 $\rightarrow$  Mean pressure layer of the 550K surface lies at about 40hPa (now mentioned in the manuscript) at all latitudes.

25. p 15, line 25: reword "..might be related to a weaker vortex."

→ done

26. p 16, line 9: insert "and [unresolved] aging by diffusion."  $\rightarrow$  done

27. p 16, line 17: Can you stress how this finding differs from past work? Citations.

→ That was mentioned in section 4.1 (p.11, line 1). However it is now repeated in the manuscript that, it was assumed in past works that diffusion makes air younger (e.g. SPARC CCM-Val-Report, 2010, Eluszkiewiecz et al.,2000, Waugh and Hall, 2002). Furthermore I mentioned additional citations.

28. p 16, line 28: Which two models? Be specific here.  $\rightarrow$  *done*

29. p. 16, line 33: insert comma: "nudging on the residual circulation[,] sensitivity studies" → done

30. p 17, line 1: insert s: "... nudging height[s] could ..."  $\rightarrow$  done

31. p 17, line 1-2: remove comma: "on the AoA trend, we conclude[,] that unresolved:"  $\rightarrow$  done

32. p 17, line 2: insert commas and reword "has a minor[,] mostly non-significant[,] impact on AoA" → done

33. p 17, line 3-5: very confusing to me, reword: "The AoA trend discrepancy between observations and both EMAC simulations still exists but cannot be explained by the trend in aging by diffusion." or something like that.  $\rightarrow$  done

34. p 17, line 6-7: What does this mean? Elaborate on "difference could not be traced back to process level." What is the process level?

 $\rightarrow$  I wanted to say that looking at the local mixing tendencies (process level) cannot explain the model differences as we look at the integrated effect, which is influenced by the residual circulation. However I decided to deleted the sentence, as it is too confusing and not clear in this context.

35. p 17, line 8: remove comma: "Note here[,] that"  $\rightarrow$  done

36. p 17, line 9: insert comma: "short data record[,] and technical"

→ done

37: p 17, line 11: insert comma: "AoA estimates[,] and AoA trends"  $\rightarrow$  done

Figure 8: Add text about the "black contours show climatological values".

→ done

[revised manuscript text omitted]

---

## Author Comment (AC2) · 24 Apr 2017

**Reply to Referee E. Ray** (ACP-2016-1144)

We thank E. Ray for the positive and constructive comments on our manuscript. Below we summarize our answers to his specific comments. Moreover, the manuscript is changed taking into account the comments. I considered all specific grammatical comments (see attached changed manuscript).

Pg. 5, line 20: 340 K is much below the tropical tropopause, why not use 380 K?
→ *That is right, to be more consistent to EMAC one should use 380K. However in CLaMS RCTTs are calculated by running backward trajectories until they cross the 340K surface (see section 3.2), as the transport through the TTL should be included in the RCTTs. Now, to be consistent with RCTTs, we use 340 K here (thus I leave the text unchanged). Note however, that the differences between EMAC and CLaMS in Figure 3 are plotted relative to the same reference layer.*

Pg. 11, line 16-18: Not sure what you are trying to say in this sentence.
→ *This sentence makes no sense, something is missing.*
*I think, I wanted to say, that local mixing tendencies are stronger in the extra tropical stratosphere (Fig. 3e), however aging by mixing differences cannot be explained here by these local mixing tendency differences, as aging by mixing is the integrated effect of the local mixing tendencies along the residual circulation. I decided to delete this sentence from the manuscript, as it is too confusing here.*

Figure 4a: The label on the x-axis, "rel. contribution to upwelling", is a bit confusing. I assume what you're showing here is the magnitude of the up-welling relative to the direct RC1SD estimate expressed as a fraction. If that's the case, the label should read more like "Relative Magnitude Compared to RC1SD direct".
→ *done*

Pg. 17, line 10: Aspects of the uncertainties in the observed AoA trend that you mention were examined in Ray et al. (2015) so that should at least be included as a reference along with Garcia et al. 2011
→ *done*

[revised manuscript text omitted]

---

## Author Response (AR2)

Oberpfaffenhofen, 11$^{th}$ Mai 2017

Dear Jayanarayanan Kuttippurath,

We thank you for your comments on ACP-2016-1144 . Now our manuscript includes a more detailed answer to the specific referee comments, you mentioned in your report. In the attached manuscript I highlighted these changes.

Concerning point " p.10, line 3-4: You assume that numerical diffusion dominates in EMAC. Is this a good assumption? Can you cite anything?":
We already answered this issue in the text (see p.10 line 11-12).

We hope, that we have now considered all points brought up by the two reviewers.

Sincerely,

Dr. Simone Dietmüller (on behalf of all co-authors)

[revised manuscript text omitted]